# Exposure to Particulate Matter as a Potential Risk Factor for Attention-Deficit/Hyperactivity Disorder in Korean Children and Adolescents (KNHANES 2008–2018)

**DOI:** 10.3390/ijerph192113966

**Published:** 2022-10-27

**Authors:** Jung-Im Shim, Garam Byun, Jong-Tae Lee

**Affiliations:** 1Department of Public Health Science, Graduate School, Korea University, Seoul 02841, Korea; 2Division of Healthcare Technology Assessment Research, National Evidence-Based Healthcare Collaborating Agency, Seoul 04933, Korea; 3Interdisciplinary Program in Precision Public Health, Korea University, Seoul 02841, Korea

**Keywords:** particulate matter, ADHD, neurodevelopmental disorder, KNHANES, child, adolescent

## Abstract

Many epidemiological studies have suggested that air pollution adversely affects neurodevelopment in children; however, evidence is still lacking. This study aimed to determine the association between particulate matter (PM) exposure and attention-deficit/hyperactivity disorder (ADHD) in children and adolescents. Data were obtained from the Korean National Health and Nutrition Examination Survey 2008–2018. Outcomes were defined from parental reports of ever doctor-diagnosed ADHD, and ADHD cases were matched to non-cases with 1:10 age–sex matching. Individual exposure levels were assigned according to each study participant’s administrative address during the year of diagnosis. Multivariate logistic regression models were used to estimate odds ratios (ORs) and 95% confidence intervals (CIs). After age–sex matching at a 1:10 ratio, the final study participants comprised 1,120 children aged 6–19 years old. A unit increase in the PM_10_ concentration was significantly associated with ADHD (OR, 1.44; 95% CI, 1.02–2.02 per 10 µg/m^3^). The association with ADHD was stronger at higher quartiles than in the lower quartiles of PM_10_ exposure; however, it was not statistically significant. Our results suggested that long-term PM_10_ exposure was associated with increased ADHD in children and adolescents. Children diagnosed with ADHD suffer from a variety of social activity and have a significant economic burden. Therefore, it is considered an important role to find out the effects of environmental risk factors, including air pollution, on children and adolescents. This may also help to increase the body of knowledge in this field and to stimulate further research.

## 1. Introduction

Attention-deficit/hyperactivity disorder (ADHD) is characterized by inattention, hyperactivity, and impulsivity. In the diagnostic criteria of the fifth edition of the Diagnostic and Statistical Manual of Mental Disorders (2013), the age of onset was changed from 7 years to 12 years, and the symptom threshold for patients aged ≥17 years was lowered from six to five symptoms [1,2].

The cause of ADHD is unclear. The development of ADHD is known to be related to the dysregulation of neurological processes caused by dopamine and norepinephrine, which are neurotransmitters in the frontal lobe of the brain [3]. In a functional magnetic resonance imaging meta-analysis study [4], hypoactivation was observed within the frontoparietal network related to executive function and the ventral attentional network, and ADHD-related dysfunction was found within multiple neuronal systems, including in the cognitive function, sensorimotor, and visual systems. In addition, the prefrontal cortex, caudate, and cerebellum are major brain areas that have a role in ADHD; these areas are interconnected by a network of neurons that control attention, thoughts, emotions, and behaviors [5].

The worldwide prevalence of ADHD in children and adolescents has been reported to be 3.4% (95% confidence interval [CI], 2.6–4.5) [6], and the community prevalence has been estimated to be between 2% and 7%, with an average of approximately 5% [7]. The prevalence in Korea differs depending on the diagnostic tool used and the region studied. In 2006, the prevalence of ADHD, estimated using the Diagnostic Interview Schedule for Children version 4, among elementary students in Seoul was approximately 13% and that among middle school and high school students was approximately 7% [8]. In another study, the prevalence in Seoul was 5.9% and 9.0% for full-syndrome ADHD and subthreshold ADHD, respectively [9]. The rate of diagnosis of ADHD in children and adolescents is higher in boys than in girls, at a ratio ranging from 2:1 to 9:1 [10]. Males are known to have higher scores in all symptom domains than females, although the severity of ADHD does not differ between boys and girls; therefore, the diagnosis is considered relatively under-recognized in girls [10,11].

ADHD is a complex and multifactorial disorder resulting from the interaction between genetic and environmental risk factors. The risk factors include maternal drinking and smoking, toxin exposure, brain damage, low birth weight, postnatal socioeconomic factors, and high heritability related to dopamine pathways [12,13]. A review of 20 twin studies [14] in the United States, Australia, Scandinavia, and Europe estimated an average heritability of 76%, making ADHD the most heritable psychiatric disorder. In addition, familial risk factors, such as parental separation or divorce, changes in primary caregivers, food insecurity, and adverse childhood experiences, contribute to the development of ADHD [12,15]. Unsupportive classrooms, peer rejection, and neighborhood environments are also considered important social interaction factors [13]. A literature review [16] has suggested various environmental pollutants, including polycyclic aromatic hydrocarbons (PAHs), black carbon, particulate matter (PM), nitrogen dioxide (NO_2_), sulfur dioxide (SO_2_), and lead (Pb), as risk factors for ADHD. According to systematic literature reviews, some studies have reported air pollution as a significant determinant of neurodevelopmental disorders; however, evidence is still limited owing to heterogeneities in the exposure window period, methods of exposure assessment, and potential confounding factors including noise, parental stress, other air pollutants, and secondhand smoke [17,18].

In Korea, only a few epidemiological studies on ADHD have been published, and evidence is insufficient on the association between air pollutant exposure in children and the development of neurodevelopmental disorders. This study aimed to investigate the relationship between PM and ADHD by using a nationally representative sample from the Korean National Health and Nutrition Examination Survey (KNHANES).

## 2. Materials and Methods

### 2.1. Data Source and Study Population

Data were obtained from the 2008–2018 (IV–VII) cycles of the KNHANES for children aged <20 years. KNHANES is a nationally representative, cross-sectional survey of the non-institutionalized civilian Korean population and is conducted with a complex, multistage, stratified, and probability cluster sampling design [19]. In 2008–2018, a total of 21,751 children aged <20 years were registered in KNHANES. We excluded children aged <6 years, those who had no parental reports, and those with other psychiatric disorders (other developmental disorders, language problems, and mental retardation). We also excluded participants with missing data on air pollution measures, covariates, or outcomes. For the case-control study design, a case was defined as a participant with parental reports of ever doctor-diagnosed ADHD. Furthermore, ADHD cases were randomly matched to non-cases by sampling without replacement through a 1:10 age–sex matching method. This study was approved by the institutional review board of the Korea university, which waved the need for informed consent because only de-identified data were used (IRB code KUIRB-2021-0003-01).

### 2.2. Exposure Assessment

PM concentration data were collected by Air Korea (www.airkorea.or.kr accessed on 21 May 2021) from region-specific government sites and were reported to the National Ambient Air Monitoring Information System. These data were confirmed and finalized by the National Institute of Environmental Research. The measurements were considered valid if the hourly data met the 75% data completeness criteria.

The study area included 16 administrative divisions and 333 monitoring stations (December 2018) across South Korea. The mean daily concentration at each monitoring site was first calculated. If >75% of the data were complete, the value was considered valid. Thereafter, we calculated the annual concentration at the monitoring sites with a missing rate of <25%. Finally, we averaged the annual means at the 16 administrative divisions.

To assign exposure measurements, the study population was restricted to children with information on the age at diagnosis of ADHD collected from parental reports. For children with ADHD, the diagnosis year was calculated as the difference between the survey year and the diagnosis age. For the control group, the diagnosis year was calculated by assuming the same diagnosis age for the sex–age matched pairs. The annual average values of PM_10_ concentration were linked to the individual administrative district codes for the year of diagnosis.

Exposure to PM_10_ was assessed using the annual mean concentration at diagnosis (lag0) and the average measurements between the year of diagnosis and the previous 4 years (lag0–1 to lag0–4). As KNHANES is a cross-sectional survey, data on the participants’ changes of address were not available. Therefore, we assumed that the participants lived in the same region during the exposure period. As exposure data are available from 2001, we excluded children diagnosed with ADHD before 2005 to account for the exposure time window.

### 2.3. Covariates

We investigated several covariates and potential confounders identified in the literature. Age, sex, body mass index (BMI) (<25 and ≥25 kg/m^2^), absence due to illness, and self-reported health (good, moderate, or bad) were used as individual factors. Socioeconomic factors included family income and insurance type (National Health Insurance or Medicaid Aid Program). Region was categorized as rural (Gyeonggi-do, Gangwon-do, Chungcheongbuk-do, Chungcheongnamdo, Jeollabuk-do, Jeollanam-do, Gyeongsangbuk-do, Gyeongsangnamdo, and Jeju-do) or urban (Seoul, Busan, Daegu, Daejeon, Gwangju, Incheon, and Ulsan). Household food security, parental stress, parental heavy drinking, and secondhand smoking were also evaluated as familial risk factors. Household food security was classified as secure or mildly insecure (food security and mild food insecurity without hunger) and moderately/severely insecure (moderate food insecurity with hunger or severe food insecurity with hunger). Parental stress was divided into low (very low or low) and high (high or very high) levels for one or both parents. Parental drinking status was also included as non-/low drinking and heavy drinking. Secondhand smoke exposure at home was defined as having any smoker in the household (yes/no) or having a parent who was a current smoker.

Ecological factors were collected from the Korean Statistical Information Service on a 16-province scale. The proportion of the elderly was calculated as the ratio of the population aged ≥65 years to the total population, and the proportion of basic livelihood security recipients was defined as the ratio of the number of recipients of basic livelihood security to the total population. The proportion of people with no high school diploma was defined as the population without a high school diploma divided by those aged 25–64 years. Education data were available for 2005, 2010, and 2015; therefore, for the 2008–2009 study population, the 2005 education data were selected. For participants in the 2010–2014 survey years, education data from 2010 were used; for those in the 2015–2018 survey years, the 2015 education data were selected.

### 2.4. Statistical Analysis

We applied a case-control design using an age–sex matching set from KNHANES. Therefore, we performed a general statistical analysis without using the SURVEY statement. Differences in basic characteristics between the two groups (ADHD vs. control) were assessed using the chi-square test for categorical variables and the independent *t*-test for continuous variables.

A logistic regression model was used to estimate the association between air pollution exposure and ADHD. The crude model included only exposure variables based on the age–sex matching set. The final model included age, sex, BMI, absence due to illness, self-reported health status, family income, type of health insurance, food security, parental stress, parental heavy drinking status, secondhand smoke exposure at home, proportion of the elderly, proportion of basic livelihood security recipients, and proportion of people with no high school diploma. The results were reported as estimated odds ratios (ORs) with 95% CIs per 10 µg/m^3^ increase in PM_10_ concentration. The effect of air pollutants on ADHD was estimated using both continuous and categorical variables, and the average exposure levels were categorized into quartiles. We conducted stratified analyses using the 1-year average PM_10_ concentration to confirm the existence of additional confounding according to age group at diagnosis, sex, region, and secondhand smoke exposure at home. All of the analyses were performed using SAS (version 9.4; SAS Inc., Cary, NC, USA) and R (version 4.1.0; R Foundation for Statistical Computing, Vienna, Austria). The statistical significance level was set at α = 0.05.

## 3. Results

Among the 13,441 children aged 6–19 years, we identified 134 ADHD cases and 13,307 controls. After age–sex matching of the ADHD cases and controls, 1,120 participants remained for analysis. The annual mean concentrations of PM_10_ for the entire country during the study period (2005–2018) are shown in Figure 1.

Table 1 shows the baseline characteristics of the ADHD cases and controls before and after matching. After matching, age and sex were balanced between the two groups. Bivariate analysis showed significant associations between ADHD and BMI, self-reported health, type of health insurance, food security, region, proportion of the elderly, proportion of basic livelihood security recipients, and proportion of people with no high school diploma.

Multivariate analysis of the various exposure time windows revealed a significant association between PM_10_ and ADHD (Table 2). For a 10 µg/m^3^ increase in the 1-year (lag0) average of the annual mean PM_10_ concentrations, the OR for ADHD was 1.44 (95% CI, 1.02–2.02). Compared with the lowest quartile of PM_10_ (Q1), the OR for ADHD in the highest quartile (Q4) was 1.81 (95% CI, 0.91–3.60). When the exposure period was longer, the increase in PM_10_ showed a positive association with ADHD; however, it was not statistically significant. For each exposure period, the association with ADHD was stronger at higher quartiles than in the lower quartiles.

Table 3 shows the association of ADHD with age, sex, region, and secondhand smoke exposure at home. The group-specific results using the 1-year average model suggested that the effect of PM_10_ exposure on ADHD was significantly stronger in girls, in participants living in urban areas, and in those with exposure to secondhand smoke (OR [95% CI] per 10 µg/m^3^: girls, 2.17 [1.00–4.72]; participants living in urban regions, 1.89 [1.10–3.25]; participants with secondhand smoke exposure at home, 1.74 [1.06–2.85]).

## 4. Discussion

This study investigated the association between PM exposure and ADHD in children and adolescents aged 6–19 years. Our results showed that an increase in PM_10_ of 10 µg/m^3^ was significantly associated with ADHD (OR, 1.44; 95% CI, 1.02–2.02). Children in the highest quartile (Q4) of exposure had a higher OR for ADHD than those in the lowest quartile (OR, 1.81; 95% CI, 0.91–3.60). This study suggests that PM influences the pathogenesis of neurodevelopmental disorders.

ADHD is a complex, highly heritable, and heterogeneous disorder. Individuals with ADHD differ from each other in different ways at many levels, such as genetic risk, environmental exposure, brain structure, and cognitive function [20]. We aimed at the effect of air pollution as environmental factors in this study.

The pathological pathway between PM exposure and ADHD remains unclear. There are two main mechanisms how PM affects the central nervous system. One is through the olfactory epithelium, directly to the brain without going through the blood–brain barrier (BBB), and the other is through the systemic circulation through the BBB to the brain [21,22]. Some evidence suggests that exposure to PM (including associated organic compounds such as PAHs or metals such as Pb, arsenic, and manganese) induces chronic inflammation and oxidative stress, which may contribute to the pathogenesis of ADHD [16,23,24,25]. Animals exposed to coarse PM (PM_10–2.5_) were shown to increase antioxidant markers (heme oxygenase 1 (HO-1) and superoxide dismutase 2 (SOD-2)) that induced physiological changes in the central nervous system and activated inflammation (interleukin 1 *beta* (IL-1β) and tumor necrosis factor-alpha(TNFα)) and unfolded protein response (X-box binding protein 1 spliced form (XBP-1S) and heat shock 70 kDa protein 5 (BiP)) [24]. Studies in Mexico City found that the brain tissue of children exposed to air pollution showed increased levels of neuroinflammatory markers and disruption of the BBB, which has been linked to neurodevelopmental disorders, including cognitive deficits [26,27]. An important mechanism of dysfunction in neurodevelopmental disorders (NDD), including autism spectrum disorder (ASD) and ADHD, is the noradrenergic system [16]. PAH, a component of PM, plays an important role as a neurotoxic factor causing disturbance of the noradrenergic system [28].

In Korea, only a few studies have investigated the relationship between air pollution and ADHD in children and adolescents. In a study using the National Health Insurance Service-National Sample Cohort (2002–2012) [29], infants (age 0–12 months) were followed-up for a 10-year period, and the risk of ADHD development due to air pollutants was 1.18 (95% CI, 1.15–1.21) for PM_10_ and 1.03 (95% CI, 1.02–1.04) for NO_2_. Another study [30] found that the short-term effects of PM_10_, NO_2_, and SO_2_ were associated with hospital admissions for ADHD. Both studies suggested that long- or short-term exposure to air pollution is associated with ADHD. Although the current study had a different design, the same association was observed.

A previous study [31] reported that short-term exposure to NO_2_ as a traffic-related air pollutant was associated with attention deficit. Children prenatally exposed to suspended PM (PM with aerodynamic diameter less than 7 µm (PM_7_)) showed delayed behavioral development such as attention, self-regulation, inhibition, and impulsivity at age 5.5 years, and it was related with ADHD in childhood [32]. In contrast, a collaborative study of eight European population-based birth/child cohorts found no evidence for the association of prenatal exposure to NO_2_ and PM with the risk of ADHD in children aged 3–10 years [33]. A Swedish twin study [34] also showed no association between pre- or postnatal exposure to traffic-related air pollution and neurodevelopmental disorders in children. Many studies have suggested the effects of air pollution factors such as PAH, black carbon, elemental carbon, benzene, and blood lead concentration, in addition to PM, NO_2_, and SO_2_, on various neurodevelopmental disorders (including ADHD) in children [16,18]. Although determining whether air pollution is the cause of ADHD is challenging, this study reports the size and consistency of the effects, similar to previous studies.

This study had several limitations. The reliance of outcomes on parental reports of doctor-diagnosed ADHD may be less reliable than a confirmed diagnosis using diagnostic tools or administrative records. The prevalence of ADHD in this study was 1.0% (134 children), which was significantly lower than the prevalence of 5.9–9.0% among children and adolescents in a previous study [9]. The general trends in KNHANES data may underestimate the prevalence collected from self-reports, because of sampling and non-sampling errors [35]. However, previous research suggested that parental reports of mental disorders in children have provided relatively reliable and accurate reporting, and were not biased to report asymptomatic conditions [36]. Therefore, large-scale prospective studies are needed for those diagnosed using the Diagnostic and Statistical Manual of Mental Disorders or parental reports of doctor-diagnosed patients.

Another important limitation is the method used to assign individual exposure levels. KNHANES provides 16 administrative codes for the residential addresses of survey participants, and variations may exist in air pollution concentrations within the districts. However, the correlations of PM_10_ concentrations between monitoring sites within each district were generally high (r > 0.69), suggesting that the variation in PM_10_ concentration in the district would not be significant. Although assigning an average exposure to each individual at a fixed monitoring site would introduce a Berkson error, the error is not expected to significantly affect the measurements and estimates [37]. To minimize measurement errors due to missing data, a concentration value was considered valid when >75% of the data were completed. In addition, we could not evaluate the effect of PM_2.5_ exposure because the data were only available from 2015. Our study subjects were from 2008 to 2018, and the number of subjects was insufficient to evaluate PM_2.5_ exposure. Further analysis of the effects of PM_2.5_ would be needed in the near future.

We also cannot completely rule out the potential residual confounding, especially due to the large spatial resolution of the data. To reduce this potential bias, the models were adjusted for several ecological variables, including the proportion of basic livelihood security recipients, proportion of the elderly, and people with no high school diploma.

Although KNHANES data include blood lead and cadmium levels, the model was difficult to analyze because of the low response rate in this study. The thyroid hormone is an essential factor for normal brain development, maturation, and function. PM and absorbed neurotoxins can potentially disrupt brain development due to dysregulation of thyroid hormone levels [38,39]. In addition, other studies have suggested noise as a potential confounding factor and reported a 5–9% significant increase in ADHD symptoms due to road traffic noise exposure [40]. As a result of the limitations of the data, it was difficult to sufficiently elucidate the effects of various pollutants on neurodevelopmental disorders. Further studies should be conducted on the effect of other neurotoxins, noise exposure, PM components, and sizes.

The strength of our study is that we considered ecological and familial risk factors as covariates using nationally representative sampling data. Ecological factors, as indicators of the sociophysical environment, were included in the model, along with individual socioeconomic factors. In addition, parental risk factors such as parental stress, secondhand smoke exposure, and heavy parental drinking were analyzed as potential confounding factors. In particular, secondhand smoke exposure has been reported to be associated with neurodevelopmental delay and poor learning ability in children and adolescents [41]. Our study analyzed a model stratified by secondhand smoke exposure, and the positive associations between PM_10_ exposure and ADHD tended to be greater in the model with secondhand smoke exposure. This suggests that environmental factors influence the risk of ADHD; however, an inverse interaction was not observed.

We applied a case-control design using age–sex matching. Because the prevalence of ADHD was different between males and females, potential confounding may have been controlled by matching. Although our results showed an association between ADHD and PM_10_ exposure in children and adolescents, the statistical significance was not consistent when comparing various exposure periods. The results of stratification analyses suggest that girls, children living in urban areas, and children who are exposed to secondhand smoke at home are more vulnerable to the effects of PM.

## 5. Conclusions

In conclusion, this study demonstrated the association between PM_10_ exposure and ADHD in children and adolescents. Our result supports the findings of previous studies on the cognitive health effects of air pollutants. Children diagnosed with ADHD suffer from problems with educational, occupational, social activity, and other psychiatric disorders, and the economic burden is significant [42,43]. In Korea, only a few epidemiological studies on ADHD have been published, and there is little evidence for the association between air pollutant exposure in children and the development of neurodevelopmental disorders. Therefore, it is meaningful in a public health perspective, in diagnosis and treatment, to examine the effects of environmental risk factors, including air pollution, on children. We suggest that it is necessary to generate sufficient evidence for the effects of air pollution and neurodevelopmental disorders.

## Figures and Tables

**Figure 1 ijerph-19-13966-f001:**
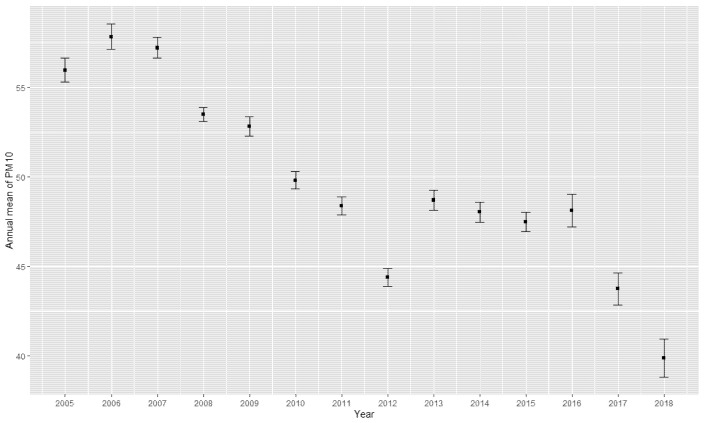
Annual mean PM_10_ concentration during the study period (2005–2018).

**Table 1 ijerph-19-13966-t001:** Descriptive characteristics of the study group aged 6–19 years from KNHANES 2008–2018.

Characteristics		Before Matching					After 1:10 Age–Sex Matching				
		ADHD Cases n = 134		Controls n = 13,307		*p*-Value	Matched ADHD Cases n = 118		Matched Controls n = 1,002		*p*-Value
Sex											
	Male	103	(76.9)	6,893	(51.8)	<0.001	90	(76.3)	780	(77.8)	0.698
	Female	31	(23.1)	6,414	(48.2)		28	(23.7)	222	(22.2)	
Age (years)											
	Mean ± SD	11.88	±3.20	11.93	±3.87	0.850	11.82	±3.17	11.37	±2.99	0.127
	6–11 years	62	(46.3)	6447	(48.5)	0.615	55	(46.6)	517	(51.6)	0.305
	12–19 years	72	(53.7)	6860	(51.6)		63	(53.4)	485	(48.4)	
Age at diagnosis (years)											
	Mean ± SD						8.09	2.81	8.37	2.76	0.297
	≤7 years						61	(51.7)	490	(48.9)	0.566
	>7 years						57	(48.3)	512	(51.1)	
BMI											
	Normal weight (<25 kg/m^2^)	116	(86.6)	12,170	(91.5)	0.045	104	(88.1)	939	(93.7)	0.024
	Overweight or obese (≥25 kg/m^2^)	18	(13.4)	1,137	(8.5)		14	(11.9)	63	(6.3)	
Absence due to illness											
	No	123	(91.8)	12,671	(95.2)	0.065	109	(92.4)	960	(95.8)	0.090
	Yes	11	(8.2)	636	(4.8)		9	(7.6)	42	(4.2)	
Self-reported health status											
	Good	70	(52.2)	9,131	(68.6)	<0.001	62	(52.5)	712	(71.1)	<0.001
	Moderate	48	(35.8)	3,572	(26.8)		45	(38.1)	249	(24.9)	
	Bad	16	(11.9)	604	(4.5)		11	(9.3)	41	(4.1)	
Household income											
	Lower	39	(29.1)	4,146	(31.2)	0.582	34	(28.8)	338	(33.7)	0.195
	Lower-middle	41	(30.6)	4,544	(34.2)		35	(29.7)	338	(33.7)	
	Upper-middle	40	(29.9)	3,342	(25.1)		38	(32.2)	234	(23.4)	
	Upper	14	(10.5)	1,275	(9.6)		11	(9.3)	92	(9.2)	
Type of health insurance											
	NHI *	124	(92.5)	12,846	(96.5)	0.012	110	(93.2)	972	(97.0)	0.032
	MAP	10	(7.5)	461	(3.5)		8	(6.8)	30	(3.0)	
Region **											
	Urban	70	(52.2)	5,922	(44.5)	0.073	64	(54.2)	399	(39.8)	0.003
	Rural	64	(47.8)	7,385	(55.5)		54	(45.8)	603	(60.2)	
Food security (familial dietary conditions)											
	Secure or mildly insecure	125	(93.3)	12,820	(96.3)	0.062	111	(94.1)	975	(97.3)	0.053
	Moderately/severely insecure	9	(6.7)	487	(3.7)		7	(5.9)	27	(2.7)	
Stress of either parents											
	Low	63	(47.0)	7,659	(57.6)	0.027	56	(47.5)	575	(57.4)	0.109
	High (one parent)	58	(43.3)	4,848	(36.4)		49	(41.5)	347	(34.6)	
	High (both parents)	13	(9.7)	800	(6.0)		13	(11.0)	80	(8.0)	
Parental drinking status											
	Low or none	94	(70.2)	10,202	(76.7)	0.076	82	(69.5)	765	(76.4)	0.101
	Heavy	40	(29.9)	3,105	(23.3)		36	(30.5)	237	(23.7)	
Secondhand smoke exposure at home											
	No	65	(48.5)	7,829	(58.8)	0.016	58	(49.2)	576	(57.5)	0.084
	Yes	69	(51.5)	5,478	(41.2)		60	(50.9)	426	(42.5)	
Proportion of the elderly											
	Mean ± SD	11.42	±2.81	11.88	±3.12	0.095	11.43	±2.77	12.51	±3.34	<0.001
Proportion of basic livelihood security recipients											
	Mean ± SD	2.73	±1.07	3.01	±1.16	0.007	2.68	±1.05	2.97	±1.2	0.011
Proportion of people with no high school diploma											
	Mean ± SD	15.76	±5.42	17.43	±6.18	<0.001	15.05	4.56	17.2	±6.44	<0.001

* The National Health Insurance covers two groups: employees (people hired by a company) and self-employed people. ** Urban areas were defined as the capital and metropolitan cities in Korea (Seoul, Busan, Daegu, Incheon, Gwangju, Daejeon, and Ulsan). Rural areas were defined as provinces in Korea (Gyeonggi, Gangwon, Chungbuk, Chungnam, Jeonbuk, Jeonnam, Gyeongbuk, Gyeongnam, and Jeju). ADHD, attention-deficit/hyperactivity disorder; SD, standard deviation; BMI, body mass index; NHI, National Health Insurance; MAP, Medical Aid Program.

**Table 2 ijerph-19-13966-t002:** Association between a 10 µg/m^3^ increase in PM_10_ concentration and ADHD according to the exposure time window of KNHANES 2008–2018 (n = 1,120).

Exposure Time Window		1:10 Age–Sex Matched			
		Crude		Adjusted	
		OR (95% CI)		OR (95% CI)	
1 year (lag0)	10 µg/m^3^	1.59	(1.21–2.07)	1.44	(1.02–2.02)
	Q1	(reference)		(reference)	
	Q2	1.01	(0.53–1.92)	1.09	(0.57–2.09)
	Q3	2.13	(1.21–3.75)	1.99	(1.08–3.66)
	Q4	2.15	(1.22–3.80)	1.81	(0.91–3.60)
2 years (lag0–1)	10 µg/m^3^	1.56	(1.19–2.05)	1.36	(0.96–1.93)
	Q1	(reference)		(reference)	
	Q2	1.59	(0.86–2.92)	1.66	(0.89–3.09)
	Q3	1.89	(1.01–3.52)	1.84	(0.96–3.51)
	Q4	2.44	(1.37–4.33)	1.79	(0.91–3.56)
3 years (lag0–2)	10 µg/m^3^	1.58	(1.20–2.07)	1.38	(0.98–1.96)
	Q1	(reference)		(reference)	
	Q2	1.22	(0.66–2.23)	1.29	(0.69–2.38)
	Q3	1.92	(1.07–3.42)	1.88	(1.02–3.46)
	Q4	2.26	(1.28–4.00)	1.83	(0.92–3.65)
4 years (lag0–3)	10 µg/m^3^	1.54	(1.18–2.00)	1.34	(0.96–1.88)
	Q1	(reference)		(reference)	
	Q2	1.20	(0.65–2.23)	1.25	(0.66–2.36)
	Q3	1.72	(0.97–3.06)	1.66	(0.89–3.09)
	Q4	2.25	(1.27–3.96)	1.75	(0.87–3.53)
5 years (lag0–4)	10 µg/m^3^	1.51	(1.17–1.95)	1.32	(0.95–1.84)
	Q1	(reference)		(reference)	
	Q2	1.40	(0.75–2.62)	1.62	(0.85–3.08)
	Q3	1.59	(0.86–2.93)	1.82	(0.94–3.54)
	Q4	2.65	(1.51–4.62)	2.22	(1.10–4.48)

Lag0: annual mean concentration at diagnosis; average lag in the diagnosis year plus 1 year before (lag0–1) to 4 years before (lag0–4). Adjusted for age, sex, body mass index, absence due to illness, self-reported health status, household income, type of health insurance, food security, stress of parents, parental heavy drinking status, secondhand smoke exposure at home, proportion of the elderly, proportion of basic livelihood security recipients, and proportion of people with no high school diploma. OR, odds ratio; CI, confidence interval; Q, quartile.

**Table 3 ijerph-19-13966-t003:** Association * between a 10 µg/m^3^ increase in PM_10_ concentration and ADHD: stratification by sex, age at diagnosis, region, and secondhand smoke exposure at home, for KNHANES 2008–2018 (n = 1,120).

	1:10 Age–Sex Matched			
	Crude		Adjusted	
	OR (95% CI)		OR (95% CI)	
All	1.59	(1.21–2.07)	1.44	(1.02–2.02)
Sex				
Male	1.55	(1.14–2.11)	1.35	(0.92–1.98)
Female	1.68	(0.97–2.91)	2.17	(1.00–4.72)
Age at diagnosis				
Age ≤7 years	1.38	(0.96–1.99)	1.42	(0.89–2.28)
Age >7 years	1.85	(1.25–2.73)	1.35	(0.83–2.21)
Region				
Urban	1.31	(0.87–1.99)	1.89	(1.10–3.25)
Rural	2.03	(1.41–2.94)	1.42	(0.80–2.51)
Secondhand smoke exposure at home				
No	1.49	(1.02–2.16)	1.24	(0.77–2.00)
Yes	1.68	(1.15–2.45)	1.74	(1.06–2.85)

* PM_10_ concentrations are assigned to the annual mean concentration at diagnosis. Adjusted for age, sex, body mass index, absence due to illness, self-reported health status, household income, type of health insurance, food security, stress of parents, parental heavy drinking status, secondhand smoke exposure at home, proportion of the elderly, proportion of basic livelihood security recipients, and proportion of people with no high school diploma. OR, odds ratio; CI, confidence interval.

## Data Availability

Not applicable.

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
