# Peer review of "Exposure to Particulate Matter as a Potential Risk Factor for Attention-Deficit/Hyperactivity Disorder in Korean Children and Adolescents (KNHANES 2008–2018)"

_ijerph, 2022, doi:10.3390/ijerph192113966_

Round 1

Reviewer 1 Report

Authors have written about an important factor in heavy metals contributing to inattention. They have taken multiple variables and the results are meaningful.

Can authors consider breaking down which pollutant contributes more and whether the clinical relevance is significant?

Can they consider speaking to whether it causes autism or ADHD? 

It may help on why this study was done? Was it done to show that it this is a finding in Korea too? If so, authors can state the hope versus asking for larger studies. 

They can revise the below sentence as it reads: these PM go into the brain.

"how A 246 common mechanism of how PM affects the central nervous system is through the olfac- 247 tory epithelium, directly into the brain without passing through the bloodbrain barrier 248 (BBB), or through the systemic circulation into the brain via the BBB [20, 21]."

Author Response

(1) Reviewer’s comment: Authors have written about an important factor in heavy metals contributing to inattention. They have taken multiple variables and the results are meaningful.

Can authors consider breaking down which pollutant contributes more and whether the clinical relevance is significant?

Author’s response: We appreciate the reviewer’s comment. We used PM10, one of the air pollution data available in Korea, and there was a limitation in not being able to evaluate the impact on other pollutants. We describe in the 'discussion' section that it is necessary to elucidate the effects of various environmental pollutants, including lead and cadmium, on neurodevelopmental disorders. We added it to the discussion. Revised discussion is shown below:

Line 321: “Due to the limitations of the data, it was difficult to sufficiently elucidate the effects of various pollutants on neurodevelopmental disorders. Further studies should be conducted to effect of various neurotoxins, noise exposure, PM components and sizes.”

(2) Reviewer’s comment: Can they consider speaking to whether it causes autism or ADHD?

Author’s response: We appreciate the reviewer’s comment. The diagnostic criteria for ADHD in ICD-10 and DSM-IV TR are attention deficit, hyperactivity, and impulsivity. Autism (ASD) includes social interaction difficulties, communication difficulties, and stereotyped behavioral disorders. Clinically, symptoms of inattention and hyperactivity are frequently reported in individuals with ASD, and children and adolescents with ADHD often suffer from problems with social interaction with their peers. However, it does not overlap with the diagnosis of ADHD and ASD [Regina T. et al., 2012]. We aimed to find the association between ADHD and PM among neurodevelopmental disorders.

Also, we could not explain the causal relationship between exposure and outcome because it was a cross-sectional study using the Korean National Health and Nutrition Examination Survey. We need further research to find a causal relationship.

Regina T., Christina S., Eva W., Michael S., Michael S., Christine F. ADHD and autism: differential diagnosis or overlapping traits? A selective review. Atten. Defic. Hyperact. Disord. 2012. 4(3), 115-39. doi: 10.1007/s12402-012-0086-2.

(3) Reviewer’s comment: It may help on why this study was done? Was it done to show that it this is a finding in Korea too? If so, authors can state the hope versus asking for larger studies.

Author’s response: We appreciate the reviewer’s comment. In Korea, only a few epidemiological studies on ADHD have been published, and there is little evidence for the association between air pollutant exposure in children and the development of neurodevelopmental disorders. This study is meaningful in a public health perspective in diagnosis and treatment by showing the association between air pollution and ADHD disease. In addition, we suggest that it is necessary to generate sufficient evidence for the effects of air pollution and neurodevelopmental disorders. We added it to the discussion. Revised discussion is shown below:

Line 349: “In Korea, only a few epidemiological studies on ADHD have been published, and there is little evidence for the association between air pollutant exposure in children and the development of neurodevelopmental disorders. Therefore, it is meaningful in a public health perspective in diagnosis and treatment to examine the effects of environmental risk factors including air pollution on children. We suggest that it is necessary to generate sufficient evidence for the effects of air pollution and neurodevelopmental disorders.”

(4) Reviewer’s comment: They can revise the below sentence as it reads: these PM go into the brain.

"how A246 common mechanism of how PM affects the central nervous system is through the olfac- 247 tory epithelium, directly into the brain without passing through the blood–brain barrier 248 (BBB), or through the systemic circulation into the brain via the BBB [20, 21]."

Author’s response: We agree with the reviewers' opinions, we revised this sentence to be clearer. Revised Discussion is shown below:  

Line 250: “There are two main mechanisms of how PM affects the central nervous system. One is through the olfactory epithelium, directly to the brain without going through the blood-brain barrier (BBB), and the other is through the systemic circulation through the BBB to the brain [21, 22].”

Reviewer 2 Report

The manuscript is relevant and interesting. I suggest the authors to be mucho more clear with the fact that differential reasons of ADDH exist, such as social contexto of development, brain maturity conditions, ways of rising and educational methods, parental participation and so on. I suggest to mention the methods of qualitative neuropsychological analysis of individual cases and their importance. Not only factor of pollution are relevant, so that there were no statistic difference in the results. This point should be stressed. This is only a possibility and, probably, more factors and reasons should be taken into account and not only the factor of pollution. From my point of view, there is a big methodological problem in research with populations, because the authors include the only one : pollution or any else. The reality is that only profound analyses of individual cases might give the right answer. 

Ii suggest to change the title, because it is only a possibility and not the reason. Possibility is not the same term as a reason.

Author Response

(1) Reviewer’s comment: The manuscript is relevant and interesting.  

I suggest the authors to be mucho more clear with the fact that differential reasons of ADDH exist, such as social contexto of development, brain maturity conditions, ways of rising and educational methods, parental participation and so on.

Author’s response: We appreciate the reviewer’s comment. We agree with the reviewers' opinions. We described in the background that ADHD is a complex and multifactorial disorder resulting from the interaction between genetic and environmental risk factors. Therefore, we considered various factors such as socioeconomic, familial, and environmental risk factor in the model within the available data. We added it to the discussion. Revised discussion is shown below:

“Line 246: ADHD is a complex, highly heritable, and heterogeneous disorder. Individuals with ADHD differ from each other in different ways at many levels, such as genetic risk, environmental exposure, brain structure and cognitive function [20]. We aimed at the effect of air pollution as environmental factors in this study.”

(2) Reviewer’s comment: I suggest to mention the methods of qualitative neuropsychological analysis of individual cases and their importance.

Author’s response: We appreciate the reviewer’s comment. This study was a cross-sectional study using the Korean National Health and Nutrition Examination Survey. The purpose of survey is to establish national health policies by examining the health and nutritional status of the people. We have limitations in using secondary data. We couldn’t examine individual neuropsychological aspects.

(3) Reviewer’s comment: Not only factor of pollution are relevant, so that there were no statistic difference in the results. This point should be stressed. This is only a possibility and, probably, more factors and reasons should be taken into account and not only the factor of pollution.

Author’s response: We appreciate the reviewer’s comment. We agree with the reviewers' opinions. To control potential confounding, we applied a case-control design using age-sex matching. Also, we adjusted for several covariates and potential confounders including individual factors (such as age, sex, BMI, self-report health), socioeconomic factor (including region, family income, insurance type), familial risk factor (including household food security, parental stress, parental drinking, secondhand smoking), and ecological factor (including proportion of the elderly, proportion of people with no high school diploma, proportion of basic livelihood security recipients).

(4) Reviewer’s comment: From my point of view, there is a big methodological problem in research with populations, because the authors include the only one : pollution or any else. The reality is that only profound analyses of individual cases might give the right answer.

Author’s response: We appreciate the reviewer’s comment, and we think it's a very good point. We aimed to evaluate PM10 as air pollutants on ADHD. We have limitations in using secondary data. We could not conduct individual analysis and had no choice but to use the method of correcting external factors for the population.

(5) Reviewer’s comment: Ii suggest to change the title, because it is only a possibility and not the reason. Possibility is not the same term as a reason.

Author’s response: We appreciate the reviewer’s comment. We suggest to change the title to:

“Exposure to particulate matter as a potential risk factor for attention-deficit/hyperactivity disorder in Korean children and adolescents (KNHANES 2008-2018)”

Reviewer 3 Report

The manuscript is  well written; clear, precise, and easy to understand.It presents the results of original research and makes a valuable contribution to knowledge and understanding of the association between particulate matter (PM) exposure and attention-deficit/hyperactivity disorder 14 (ADHD) in children and adolescents. However I see the following minor issues that should be resolved before publishing this manuscript;

Line 266 :  Children prenatally exposed to PM7, NO2, and SO2 showed delayed behavioral development in a nationwide population-based longitudinal survey in Japan [30]. It was identified that PM7 showed delayed behavioral development. Please check the data of reference  [29] and [30]. 

Author Response

Reviewer’s comment:

The manuscript is well written; clear, precise, and easy to understand. It presents the results of original research and makes a valuable contribution to knowledge and understanding of the association between particulate matter (PM) exposure and attention-deficit/hyperactivity disorder 14 (ADHD) in children and adolescents. However, I see the following minor issues that should be resolved before publishing this manuscript;

Line 266:  Children prenatally exposed to PM7, NO2, and SO2 showed delayed behavioral development in a nationwide population-based longitudinal survey in Japan [30]. It was identified that PM7 showed delayed behavioral development. Please check the data of reference [29] and [30].

Author’s response: We appreciate the reviewer’s comment. We reviewed the literature extensively on the effects of air pollution on attention deficit and neurodevelopment related to ADHD. Therefore, we agree with the reviewers' opinions, revised this sentence to the Discussion section. Revised Discussion is shown below:  

Line 277: “Children prenatally exposed to suspended PM (PM with aerodynamic diameter less than 7 µm (PM7) showed delayed behavioral development like attention, self-regulation, inhibition and impulsivity at age 5.5 years, and it was related with ADHD in childhood [32].”

Reviewer 4 Report

Please add more information about patophysiological relations of air pollution by PM10 and hyperactivity disorder.

Also, I recommend to change title of the paper to "Association of environmental factors and attention-deficit/hyperactivity disorder in children".  As you evaluating more factors, not only PM. 

Also please answer why you choose PM10, not PM2,5 or ather pollutants? PM10 is not going deeply into organism, do not have very big influence to health.

Author Response

(1) Reviewer’s comment: Please add more information about pathophysiological relations of air pollution by PM10 and hyperactivity disorder.

Author’s response: We appreciate the reviewer’s comment. We added this information to the Discussion section. Revised Discussion is shown below:

Line 257 “Animals exposed to coarse PM (PM2.5-PM10) were shown to increase antioxidant markers (HO-1 and SOD-2) that induced physiological changes in the central nervous system and activated inflammation (IL-1β and TNFα) and unfolded protein response (XBP-1S and BiP) [24]. Studies in Mexico City found that the brain tissue of children exposed to air pollution showed increased levels of neuroinflammatory markers and disruption of the BBB, which has been linked to neurodevelopmental disorders including cognitive deficits [26,27]. An important mechanism of dysfunction in neurodevelopmental disorders (NDD), including autism spectrum disorder (ASD) and ADHD, is the noradrenergic system [16]. PAH, a component of PM, play an important role as a neurotoxic factor causing disturbance of the noradrenergic system [28].”

(2) Reviewer’s comment: Also, I recommend to change title of the paper to "Association of environmental factors and attention-deficit/hyperactivity disorder in children".  As you evaluating more factors, not only PM.

Author’s response: We appreciate the reviewer’s comment and we think it's a very good point. We aimed to evaluate PM10 among environmental factors. Socioeconomic factors such as region and second-hand smoking at home, and parent risk factor were considered as confounding or effect modifier of PM. We would like to change the title to:

“Exposure to particulate matter as a potential risk factor for attention-deficit/hyperactivity disorder in Korean children and adolescents (KNHANES 2008-2018)”

(3) Reviewer’s comment: Also please answer why you choose PM10, not PM2,5 or ather pollutants? PM10 is not going deeply into organism, do not have very big influence to health.

Author’s response: We agree with the reviewers' opinion, however we could not be applied PM2.5 exposure because the data were established in 2015. Our study subjects were from 2008 to 2018, and the number of subjects was insufficient to evaluate PM2.5 exposures.

Ultrafine PM(PM2.5) is greater effects than PM10 because of the small size of PM, but inhalable PM is a mixture of particles suspended in the air [Ran Y. et al., 2022]. PM contains organic components like PAHs, Pb, As and Mn, which still carries the risk of causing neurotoxicity.  Therefore, our study aimed to the association ADHD and PM10 as a traffic-related air pollution.

Ran Y., Yuen-Shan H., Raymond C.-C.C. The pathogenic effects of particulate matter on neurodegeneration: a review. J. Biomed. Sci. 2022. 29(1), 15. doi: 10.1186/s12929-022-00799-x.

Round 2

Reviewer 4 Report

Can be published.

Author Response

Responses to the Associate Editor’s and Reviewers’ Comments

20 October, 2022

Dear reviewers and editorial staffs in International Journal of Environmental Research and Public Health

We are sincerely grateful for your thorough consideration and scrutiny of our manuscript, “Association between exposure to particulate matter and attention-deficit/hyperactivity disorder in children and adolescents”, control number ijerph-1887658. Through the accurate comments made by the reviewers, we better understand the critical issues in this paper. We have revised the manuscript according to the Reviewer’s suggestions. We hope that our revised manuscript will be considered and accepted for publication in the International Journal of Environmental Research and Public Health. We acknowledge that the scientific and clinical quality of our manuscript was improved by the scrutinizing efforts of the reviewers and editors.

The changes within the revised manuscript were highlighted (underlined and in blue). Point-by-point responses to the reviewers’ comments are provided below.

<MINOR COMMENTS>

 (1) Reviewer’s comment: I appreciate that the authors have been generally responsive to the reviewers' comments. I have some minor comments.

  1. The authors have responded to change the title of their paper, according to the recommendation of Reviewers 2 and 4. However, the title of the latest version has not been changed.

Author’s response: We appreciate the reviewer’s comment. We changed the title of our article in this version.

“Exposure to particulate matter as a potential risk factor for attention-deficit/hyperactivity disorder in Korean children and adolescents (KNHANES 2008-2018)”

(2) Reviewer’s comment: 2. The reason that the effects of PM2.5 could not be evaluated in this study should be mentioned as a limitation.

Author’s response: We appreciate the reviewer’s comment. We think it's a very good point, and we agree with the reviewer's opinion. We added it to the discussion. Revised discussion is shown below:

Line 312: “In addition, we could not be applied PM2.5 exposures because the data were established in 2015. Our study subjects were from 2008 to 2018, and the number of subjects was insufficient to evaluate PM2.5 exposures. More detailed exposure data and further analysis of the effects of PM2.5 are needed.”

(3) Reviewer’s comment: 3. Line 261. I recommend "PM10-2.5" as an abbreviation of coarse PM

Author’s response: We appreciate the reviewer’s comment. We added it to the discussion. Revised discussion is shown below:

Line 259: “Animals exposed to coarse PM (PM2.5-10) were shown to increase antioxidant markers (HO-1 and SOD-2) that induced physiological changes in the central nervous system and activated inflammation (IL-1β and TNFα) and unfolded protein response (XBP-1S and BiP) [24].”
